# N-Way NIR Data Treatment through PARAFAC in the Evaluation of Protective Effect of Antioxidants in Soybean Oil

**DOI:** 10.3390/molecules25194366

**Published:** 2020-09-23

**Authors:** Larissa Naida Rosa, Thays Raphaela Gonçalves, Sandra T. M. Gomes, Makoto Matsushita, Rhayanna Priscila Gonçalves, Paulo Henrique Março, Patrícia Valderrama

**Affiliations:** 1Universidade Estadual de Maringá (UEM), Maringá, Paraná 87320-900, Brazil; larissanaida@gmail.com (L.N.R.); thays_rapha@hotmail.com (T.R.G.); stmg@uem.br (S.T.M.G.); mmakoto@uem.br (M.M.); 2Universidade Tecnol·ógica Federal do Paraná (UTFPR), Campo Mourão, Paraná 87301-899, Brazil; rhayanna_pg@hotmail.com (R.P.G.); paulohmarco@utfpr.edu.br (P.H.M.)

**Keywords:** NIR spectroscopy, chemometric, PARAFAC, oil, antioxidants, oxidation products, thermal degradation, plant-based extract, correlation map

## Abstract

The use of chemometric tools is progressing to scientific areas where analytical chemistry is present, such as food science. In analytical food evaluation, oils represent an important field, allowing the exploration of the antioxidant effects of herbs and seeds. However, traditional methodologies have some drawbacks which must be overcome, such as being time-consuming, requiring sample preparation, the use of solvents/reagents, and the generation of toxic waste. The objective of this study is to evaluate the protective effect provided by plant-based substances (directly, or as extracts), including pumpkin seeds, poppy seeds, dehydrated goji berry, and Provençal herbs, against the oxidation of antioxidant-free soybean oil. Synthetic antioxidants tert-butylhydroquinone and butylated hydroxytoluene were also considered. The evaluation was made through thermal degradation of soybean oil at different temperatures, and near-infrared spectroscopy was employed in an n-way mode, coupled with Parallel Factor Analysis (PARAFAC) to extract nontrivial information. The results for PARAFAC indicated that factor 1 shows oxidation product information, while factor 2 presents results regarding the antioxidant effect. The plant-based extract was more effective in improving the frying stability of soybean oil. It was also possible to observe that while the oxidation product concentration increased, the antioxidant concentration decreased as the temperature increased. The proposed method is shown to be a simple and fast way to obtain information on the protective effects of antioxidant additives in edible oils, and has an encouraging potential for use in other applications.

## 1. Introduction

The last 50 years were marked by technological advancements in the instrumental area, which reached all science fields, and especially chemistry [1]. With this new scenario, analytical chemistry has taken on new characteristics, and tools based on chemometrics are starting to be used in order to extract information from the big data generated in analytical chemistry [2,3,4,5,6,7]. Through time, chemometrics continues to evolve in the field of analytical chemistry [8,9,10] and is advancing to other scientific areas where analytical chemistry is highly present, such as food science [11].

In the food analysis, edible oils are submitted to exploration in a wide range of possibilities, such as authentication [12], oxidation state [13], adulteration [14], and so on. Moreover, there are several studies on the search to explore the herbs and seeds’ influences on the protection/oxidation of edible oils [15,16,17,18,19,20]. However, it was observed that this kind of study is still performed by time-consuming analytical methods, requiring sample preparation, which demands the use of solvents/reagents, therefore generating potential toxic waste. By considering this fact, the development of methodologies which allow fast and clean analyses to be made is a real priority, even more so in a world in which diseases, which can also be caused or spread by food consumption, are causing concern among consumers worldwide. One suitable method draws upon near-infrared (NIR) spectroscopy, which has proven to be effective for food analysis [21], particularly in the omics era [22].

In this sense, the main objective of this study was to evaluate the protective effect provided by plant-based substances in which the phenolic content and antioxidant activity were previously described in the literature [19,23,24,25], i.e., hydroalcoholics of plant-based extracts of pumpkin (*Cucurbita* sp.) seeds (PPSE), poppy (*Papaver somniferum* L.) seeds (PSE), dehydrated goji berry (*Lycium barbarum* L.) (DGBE), and Provençal herbs (a cocktail of four herbs: rosemary—*Salvia rosmarinus* L.; sage—*Salvia officinalis* L., thyme—*Thymus vulgaris* L., and oregano—*Origanum vulgare* L.) (PHE). These extracts were added to soybean oil to check for the (anti)oxidation effect in comparison to antioxidant-free soybean oil (AFSO—or the control sample). The effect was also studied of adding pumpkin seeds (PPS), poppy seeds (PS), dehydrated goji berry (DGB), and Provençal herbs (PH) directly to the AFSO. From this, PPSE, PSE, DGBE, PHE, PPS, PS, DGB, and PH were denominated plant-based substances with antioxidant properties. Considering the known antioxidant properties of these plant-based substances, a protective effect was expected. The results were also compared to AFSO with added synthetic antioxidants tert-butylhydroquinone (TBHQ) and butylated hydroxytoluene (BHT) through the thermal degradation of soybean oil at different temperatures. This process was monitored by NIR spectroscopy, in an n-way mode, coupled with the chemometric tool, Parallel Factor Analysis (PARAFAC).

Phenolic compounds in herbs are the major components responsible for scavenging radical species by donating a hydrogen atom or an electron to form stable compounds [18]. Furthermore, studies on the medicinal properties of herbs show the contribution of the phytochemical constituents, namely polyphenols that have antioxidant properties [23]. Regarding goji berry and the seeds employed in this work, the goji berry is a raw material for the extraction of phenolic compounds that is added to food products to protect against oxidation [19]. The main phenolic compounds in pumpkin seed oil are tyrosol, vanillic acid, caffeic acid, o-coumaric acid, besides some small amounts of trans-cinnamic acid. Beyond phenolic acids, daidzein and genistein are also present in pumpkin seed oil. From the lignan group of phenolics, secoisolariciresinol was determined in significant amounts in the seeds [24]. On the other hand, polyphenols are reported to be present in poppy seeds [25].

The PARAFAC was applied to NIR spectroscopy data in an n-way mode. This approach was previously reported for the evaluation of rice oil from different countries, subjected to thermal degradation [26]. To apply PARAFAC, it is mandatory to acquire one response matrix for each sample, so it can be organized in a tensor form. In this case, each response matrix was obtained from soybean oil (with the various plant-based antioxidants added) which was exposed to different temperatures. So, the final matrix for each oil was composed of 9 rows, referring to the 9 different temperatures, and 125 variables, related to the wavelengths measured by the equipment.

## 2. Results and Discussion

Oxidation products occur in several edible oils when they are submitted to heat from ambient to high temperatures, such as during frying [13,27,28,29]. According to Gonçalves et al. [13], soybean oil starts to be oxidized at a temperature of about 50 °C [13]. The antioxidant properties of the plant-based substances considered in this study have been previously reported. In the literature, it has been noted that pumpkin seeds are rich in phytosterols [30], that goji berries include phenolic acids and flavonoids, and due to their antioxidant properties, are used for the development of drugs, cosmetics, and special purpose foods [19], that poppy seeds contain polyphenolic antioxidants [25,31], and that provençal herbs, i.e., a cocktail of rosemary, sage, thyme, and oregano, have antioxidant effects, their main antioxidants being carnosol, carnosic acid, carvacrol, and thymol [32]. Several studies have been undertaken using NIR spectroscopy, as opposed to the official Rancimat method, as a technique to monitor oxidative stability, with the two methods showing high correlation and equivalent precision [33,34,35].

Regarding the usage of chemometric tools in the oxidation process of edible oils, the effect of a biological antioxidant added to sunflower oil has been previously reported, describing the usage of fluorescence spectroscopy and independent components analysis [36]. Also, the thermal degradation of rice oil was evaluated by ultraviolet and visible (UV–Vis) spectroscopy and NIR, with the data being organized in three-way arrays for evaluation by PARAFAC [26]. Moreover, the thermal degradation of colza, soybean, corn, sunflower, and olive oils has been evaluated by UV-Vis spectroscopy and multivariate curve resolution with alternating least squares (MCR-ALS) [13]. Multiway decomposition of the NIR data in a three-way array can be achieved by using PARAFAC, which allows for the evaluation of the protective effect of plant-based substances with antioxidant properties against the oxidation of soybean oil. It is important to note that one of the main reasons for choosing soybean oil in this study was related to the statistics of its production and consumption in Brazil. According to a study performed by the Brazilian Association of Vegetable Oil Industries, in 2019, a total of 8.791 tons of soybean oil were produced in Brazil, and from this, a total of 7.909 tons were used domestically [37].

The NIR spectra of the heated samples (Figure 1) were preprocessed through the Savitzky Golay algorithm [38], by applying the first derivative (first order polynomial applied within a sliding three-point spectral window). It is possible to observe very slight differences between samples, mainly related to the absorbance intensities. However, considering the lack of selectivity in NIR spectroscopy it is hard to draw conclusions about thermal oxidation in oil samples added with different substances/antioxidants, and compare which one can promote a more pronounced protective effect.

The data were organized in an n-way mode (Figure 2) assembled with eleven samples (antioxidant-free soybean oil or control, and antioxidant-free soybean oil added with BHT, TBHQ, Provençal herbs extract, dehydrated goji berry extract, pumpkin seed extract, poppy seed extract, Provençal herbs directly, dehydrated goji berry directly, pumpkin seed directly, poppy seed directly), nine different temperatures and 125 wavelengths.

Within the contour maps (i.e., bidimensional matrix representation of a sample heated in different temperatures and monitored with NIR spectra), more spectral differences between samples can be visualized. It is possible to observe the differences between them due to the addition of plant-based substances (directly or extracts) or synthetic antioxidants. These differences occur with the temperature (abscissa-axis), wavelength (ordered-axis), and absorbance intensity (color bar), but even in this way is hard to conclude only visually.

To improve the visualization and interpretation of this second-order multivariate data set (i.e., n-way data), PARAFAC was applied. This chemometric tool is a second-order method that provides unique solutions, which has great advantages for modeling spectroscopic data. For trilinear data sets, the number of components is equivalent to the number of active chemical species presented in the sample. In addition, PARAFAC provides a one score matrix and two loading matrices [39].

The first step to apply PARAFAC is the choice of an appropriate number of factors. Normally, in complex matrices, the number of components is unknown and must be determined from the data using mathematical diagnostic tools [40], such as a core consistency diagnostic or the “corcondia” value [41]; the latter was chose for use in this study.

The PARAFAC model for the three-way array was performed using two factors with no constraints, where the corcondia value result was 100%, implying an appropriate dimensionality [41]. The PARAFAC results are presented in Figure 3, where the scores contain information about the samples. One loading mode is related to the temperature and, as a consequence, resembles the kinetics profiles, i.e., changes occur depending on the nature of the component responsible for the factor, evolving with increasing temperature. The second loading mode is related to the spectra and reveals the pure spectrum of the component responsible for the factor.

The PARAFAC scores on factor 1 (Figure 3A) show the behavior performed for plant-based substances compared to TBHQ, BHT, and control samples. The results suggest that control, TBHQ, PPSE, PSE, DGB, PPS, and PS present similar behavior in the oxidation process, while BHT and PHE are inferior, and DGBE and PH are more delayed in this process. The loading profile related to the temperature on factor 1 (Figure 3B) displays an increasing evolution, which is a characteristic of oxidation products, according to previous studies [13,28,29]. Even more, based on the loadings from the temperature mode, the concentration of the oxidation products starts to increase at 50 °C, which agrees with a previous study performed with commercial soybean oil [13].

Considering the PARAFAC loadings for the spectral mode on factor 1 (Figure 3C), there is agreement with the correlation analysis between the UV absorptions at 210 and 270 nm and the NIR spectra of the samples heated at 150 °C (Figure 3D). The primary compounds of oxidation exhibit maximum absorbance in the UV region at 220–234 nm [41,42]. The samples were monitored at 230 nm when heated at 150 °C. As a consequence of the heating process, the formation of trienes and unsaturated ketones or aldehydes, i.e., secondary products of oxidation that show absorption at 270 nm [42,43], also takes place. Based on the correlation results, all the red parts of the spectra in Figure 3D have a positive correlation with primary and secondary oxidation products, being the region from 1190 to 1210 more strongly correlated with oxidation products (correlation close to 1).

Hydroperoxide evaluation through NIR spectroscopy was reported in a study from 1958 [44] showing absorption at 1420 nm and 1450 nm. According to Wójcicki et al. [45], the pronounced changes in the NIR range during oxidation involved a decrease of the band intensity at around 1700 nm, identified as the first overtone of the C-H in the methylene group, and the second overtone band of the methylene group at around 1165 nm. A vibration band at around 1440 nm that increased in intensity was assigned to peroxides. Based on these reports [44,45], the NIR regions assigned to hydroperoxides were negatively correlated with UV absorptions at 210 and 270 nm.

To the second PARAFAC factor, the information provided regards the antioxidant properties of plant-based substances, TBHQ, BHT, and control samples. According to Rosa et al. [46], the NIR absorptions below 1000 nm, from 1200 to 1350 nm, and above 1400 nm is attributed to the samples of edible oils that demonstrated the lower amount of tocopherol and phenolic compounds. Based on this, the scores on factor 2 (Figure 3E) reveal an equivalence between BHT, TBHQ, and plant-based substances added directly to the control sample. The extract of the plant-based substances displayed excellent antioxidant potential: PHE, PPSE, and PSE were equivalent, while DGBE was less effective than the others.

The loadings plot from the temperature mode on factor 2 (Figure 3F) agrees with several studies [13,28,29,36], that show the antioxidant concentrations decrease over time. The spectral mode for the PARAFAC loadings on factor 2 (Figure 3G) presents some similarities with the tocopherol NIR spectra [46]. However, considering that the tocopherol content in vegetable oils is directly related to the type of processing applied, and that, with chemically refined oils, the content of this constituent is reduced by up to 80% [47], in this case, the recovered spectra suggest the absorption of antioxidants as a whole, i.e., tocopherol from soybean oil plus the added substances with antioxidant properties.

From this proposal, the achieved results suggest that multiway decomposition of the NIR data in a three-way array using PARAFAC is a feasible tool for the evaluation of the protective effect of antioxidants against the deterioration of soybean oil, since traditional analytical methods have previously shown antioxidant properties and phenolic content in plant-based substance extract, as used in this work. For this reason, traditional analytical methods were not applied. Nonetheless, in the study by Andjelkovic et al. [24], the total phenolics content in the pumpkin seed oil ranged from 25 to 51/mg gallic acid equivalent (GAE)/kg of oil. The main phenolics were tyrosol, vanillic acid, vanillin, luteolin, and sinapic acid, while the maximum antioxidant capacity measured by the reduction of the 2,2-diphenyl-1-picrylhydrazyl (DPPH) radical was 62%. The addition of pumpkin seed oil was reported as improving the frying stability of canola oil, and this antioxidative effect was attributed to its phenolic composition [48].

The methanolic extracts of Poppy seed presented a total phenolics content of 1937.7 mg GAE/100 g extract, and the antioxidant capacity estimated by the reduction of the DPPH radical ranged from 44 to 66.5% [25]. Goji berry ethanolic extracts presented total phenolics of 1351.45 mg GAE/100 g extract, and antioxidant activity by DPPH radical of 1.64 millimoles of Trolox equivalent (mmol TE)/100 g of dry weight of sample [19]. Rosemary, oregano, sage, and thyme, the constituents of Provençal herbs, present respectively 83, 33.5, 192, and 33 mmol TE/g dry weight of sample [49].

## 3. Materials and Methods

### 3.1. Samples

Poppy seeds, pumpkin seeds, dehydrated goji berry, and Provençal herbs used in this study were acquired in Compiègne, France. The refined antioxidant-free soybean oil, as well as the synthetic antioxidants BHT and TBHQ, were provided by an edible oil industry located in Paraná State, in the South of Brazil.

### 3.2. Sample Preparation by Using Plant-Based Substances Directly

The addition of the plant-based substances directly to the antioxidant-free soybean oil samples was made following the methodology described by Soares et al. [50]. Each plant-based substance was crushed, and 5% (*w/v*) of this one was added, individually, in the antioxidant-free soybean oil. The quantities of synthetic antioxidants TBHQ and BHT also followed the same methodology, being incorporated 5% (*w/v*) in the antioxidant-free soybean oil. Each sample was mixed for 10 min and then filtrated by using a vacuum system.

### 3.3. Sample Preparation by Using Plant-Based Substance Extracts

The plant-based substance extracts were prepared according to a method described by Laczkowski et al. [51]. For each sample, 1 g of the plant-based material was crushed and added to a 10 mL of ethanol/water solution 70:30 (*v/v*). The extraction was performed by using ultrasound (Fischer Scientific—FB120 Model) with a probe (Fischer Scientific—E18 Model), 120 W power, and 100% frequency (20 kHz), performed at room temperature for 15 min. The obtained extract was centrifuged for 20 min at 400 rpm. The hydroalcoholic extracts were added at a concentration of 5% (*v/v*) to the antioxidant-free soybean oil and mixed for 10 min.

### 3.4. Thermal Degradation

Samples (control, with synthetic antioxidants, and with the addition of plant-based substances) were heated from 30 to 170 °C, increasing by steps of 20 °C; the first evaluation was made at room temperature (25 °C). The samples were heated in a glass beaker using a heating plate with stirring. The experiments were conducted in an open system, simulating conventional frying. All samples were heated, and aliquots were collected at nine different temperatures (25, 30, 50, 70, 90, 110, 130, 150, and 170 °C). These aliquots were put to cool at room temperature and NIR spectra were acquired after samples reached stable 25 °C for 30 min.

### 3.5. Spectroscopy Measurements and Data Treatments

Samples were directly analyzed using a handheld dispersive NIR spectrometer (microNIR™ 1700, JDSU Uniphase Corporation, Milpitas, CA, USA) from JDSU Uniphase Corporation with a glass cuvette (1 cm diameter) in absorbance mode. The absorbance spectra of each sample were recorded from 900 to 1680 nm, with 32 scans and 6.00 nm resolution. The equipment standardization was performed as suggested by the manufacturer, where a diffuse reflectance pattern of 99% was used as blank, while the dark (zero) reflectance was standardized with the lamp off, as suggested by the manufacturer.

Data were analyzed using MATLAB version R2007b (The Mathworks Inc., Natick, MA, USA) where the chemometric evaluation was performed by PARAFAC with calculations carried out by using the N-way toolbox for MATLAB version 3.1 [52]. Mathematical steps for the PARAFAC tool are detailed described by Bro [36].

UV measurements were acquired at 230 and 270 nm in a 1 mm quartz cuvette, by using an Ocean Optics equipment (Ocean Optics, Largo, FL, USA). These data were acquired for each oil sample at the temperature of 150 °C (Appendix A) and employed in a correlation with the NIR spectra of samples heated at the same temperature. This was the maximum temperature in which the UV measurement from some samples was possible, especially those added with plant-based substances. The correlation analysis was made, and mathematical descriptions are presented in Vilas-Boas et al. [53].

## 4. Conclusions

Soybean oil with added plant-based substances and synthetic antioxidants was evaluated by NIR spectroscopy in three-way arrays (samples × temperatures × absorbance at different wavelengths) by applying the PARAFAC chemometric tool to evaluate the protective effect against oxidation, making it possible to determine the behavior of the oil when it was submitted to heating.

Changes and evolution in the spectral profiles were evaluated during heating; the oxidation product concentration increased while antioxidant concentration decreased as the temperature increased. These observations allowed us to conclude that the plant-based extract was more effective at improving the frying stability of soybean oil.

The applied method was shown to be a simple and fast way to obtain information about the protective effect of synthetic antioxidants and plant-based substances added to edible oils.

## Figures and Tables

**Figure 1 molecules-25-04366-f001:**
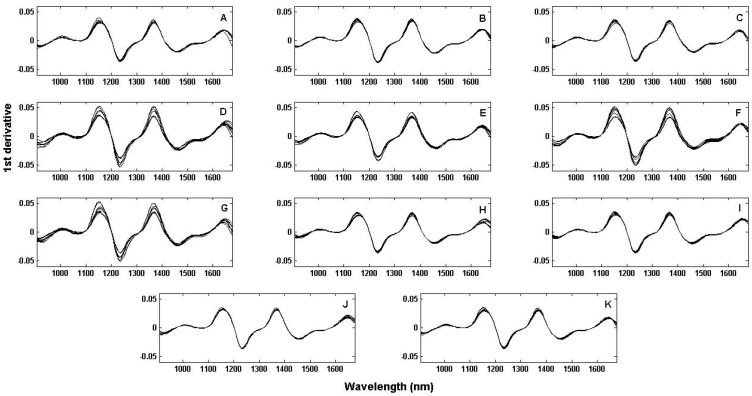
NIR spectra: (**A**) Control (i.e., antioxidant-free soybean oil); (**B**) BHT; (**C**) TBHQ; (**D**) Provençal herbs extract; (**E**) Dehydrated goji berry extract; (**F**) Pumpkin seed extract; (**G**) Poppy seed extract; (**H**) Provençal herbs; (**I**) Dehydrated goji berry; (**J**) Pumpkin seed; (**K**) Poppy seed.

**Figure 2 molecules-25-04366-f002:**
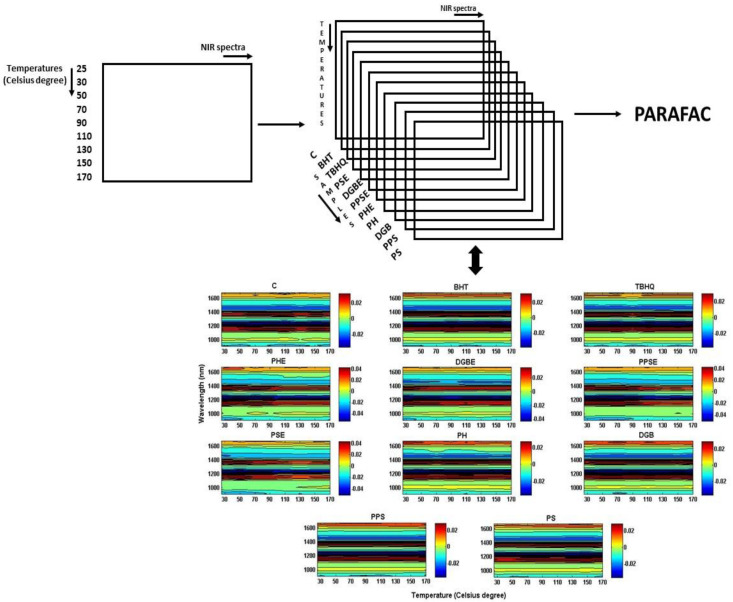
Scheme for illustration of the NIR spectra organization in n-way mode. C = Control (i.e., antioxidant-free soybean oil). BHT = butylated hydroxytoluene. TBHQ = tert-butylhydroquinone. PSE = Poppy seed extract. DGBE = Dehydrated goji berry extract. PPSE = Pumpkin seed extract. PHE = Provençal herbs extract. PH = Provençal herbs. DGB = Dehydrated goji berry. PPS = Pumpkin seed. PS = Poppy seed.

**Figure 3 molecules-25-04366-f003:**
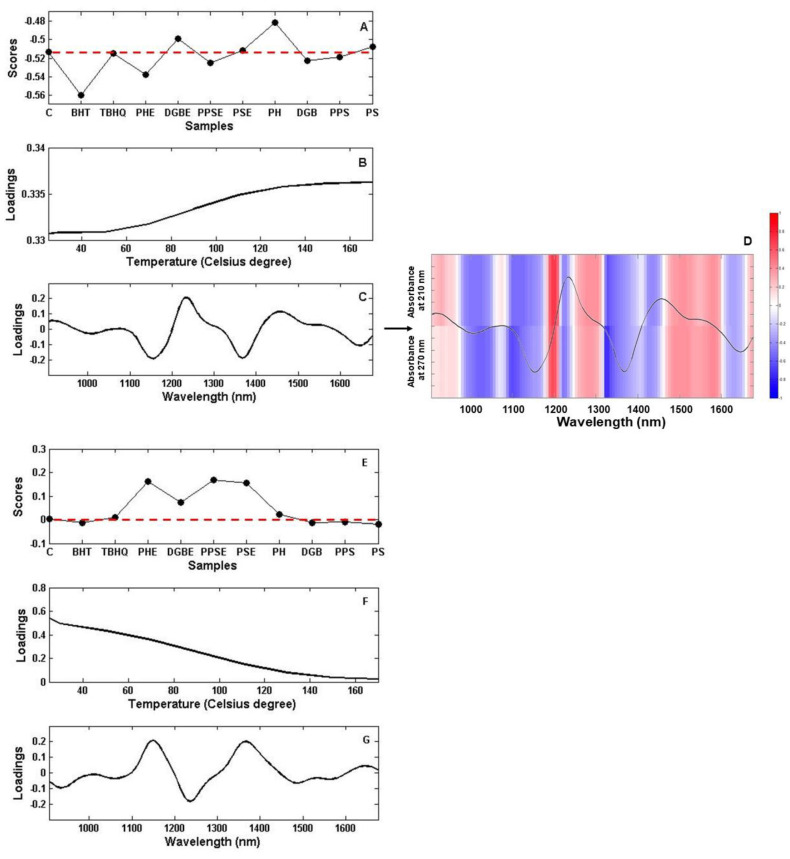
PARAFAC results. (**A)** Scores on factor 1. (**B**) Loadings on factor 1 related to temperature. (**C**) Loadings on factor 1 related to spectral mode. (**D**) Correlation map through samples relating the NIR spectra at 150 °C versus the ultraviolet univariate result. (**E**) Scores on factor 2. (**F**) Loadings on factor 2 related to temperature. (**G**) Loadings on factor 2 related to spectral mode. C = Control (i.e., antioxidant-free soybean oil). BHT = butylated hydroxytoluene. TBHQ = tert-butylhydroquinone. PSE = Poppy seed extract. DGBE = Dehydrated goji berry extract. PPSE = Pumpkin seed extract. PHE = Provençal herbs extract. PH = Provençal herbs. DGB = Dehydrated goji berry. PPS = Pumpkin seed. PS = Poppy seed.

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
