# Peer review of "N-Way NIR Data Treatment through PARAFAC in the Evaluation of Protective Effect of Antioxidants in Soybean Oil"

_molecules, 2020, doi:10.3390/molecules25194366_

Round 1

Reviewer 1 Report

Please justify the choice of plant products in this study and complete them with chemical data (in the Introduction chapter). Is there a correlation between the chemical composition of each medicinal species and its antioxidant action?
Also, why was soybean oil chosen? Does it oxidize more easily? Could the olive oil have been tested? Olive oil is perhaps the most used by the population.
The plant species used in this study should be identified by a qualified person (botanist, biologist) and each plant should contain an identification number (voucher number). The antioxidant action, as well as the determination of polyphenols should have been made on the extracts obtained by the authors. In this case, the results would have been more accurate.

Author Response

Dear Reviewer,

            Thank you for your attention to the manuscript molecules-934375 entitle “N-way NIR data treatment through PARAFAC in the evaluation of antioxidants protective effect in soybean oil”.

The reviewers' comments were evaluated, and the suggestions were accepted (highlighted in red on the text). We thank the reviewers for their suggestions and contributions. We hope that all these changes have contributed to improving the overall quality of the paper. We also hope that it now meets the publication standards in Molecules.

Thank you for considering this reviewed manuscript.

Sincerely,

Prof. Patrícia Valderrama

Please justify the choice of plant products in this study and complete them with chemical data (in the Introduction chapter).

We agree. The introduction chapter was improved to include this suggestion.

Is there a correlation between the chemical composition of each medicinal species and its antioxidant action?

Based on the literature the polyphenols are the main constituents responsible for the antioxidant action. This information was included in the introduction chapter: “…studies on the medicinal properties of herbs show the contribution of the phytochemical constituents, namely polyphenols that have antioxidant properties [21].” Furthermore, literature references reporting the polyphenols in goji berry, pumpkin, and poppy seeds were included in the introduction chapter also.

Also, why was soybean oil chosen? Does it oxidize more easily? Could the olive oil have been tested? Olive oil is perhaps the most used by the population.

An explanation was included in the manuscript: “…the main goal to choose soybean oil in this study considered the statistics of production and domestic consumption of soybean oil in Brazil. According to a study performed to the Brazilian Association of vegetable oil industries, in 2019 a total of 8.791 t of soybean oil was produced in Brazil, and from this amount, a total of 7.909 t was used in domestic consumption [35].” Moreover, based on a previous study with colza, soybean, corn, sunflower, and olive oils showed that soybean oil starts its oxidation products concentrations increase starting at 50 °C. This information was highlighted in the manuscript.

The plant species used in this study should be identified by a qualified person (botanist, biologist) and each plant should contain an identification number (voucher number).

Unfortunately, we talked with the professor biologist responsible for this catalog action here in our university, and he recognizes the importance of this voucher number when the authors cultivate the plant. In our case, we utilize commercial samples of plant parts and previous work in this sense (with commercial samples from plant parts added to oil – cited in the reference list as a number from 15 to 20) also don't have a voucher number.

The antioxidant action, as well as the determination of polyphenols should have been made on the extracts obtained by the authors. In this case, the results would have been more accurate.

We agree that these determinations could become our results more reliable. However, there are several studies on literature that report these results. We present justification in the manuscript for not realize these analyses and discuss the results achieved by previously published works.

Reviewer 2 Report

In manuscript molecules-934375, N-way NIR data treatment through PARAFAC in the evaluation of antioxidants protective effect in soybean oil was investigated, which suggested a simple and fast way to obtain information about the protective effect of synthetic antioxidants and plant-based substances added in edible oils. The methods and results were clear and interesting. I suggest accept is after minor revision.

1 The method of ‘NIR spectroscopy in three-way arrays (samples x temperatures x absorbance at different wavelengths) by applying the PARAFAC chemometric tool to evaluate the protective effect over oxidation’ is interesting. Whether this method was applied in other components evaluation? Whether have other research reported this method? I suggest add the introduction of principle and application of methods in Part introduction.

2 In all figures, what is control? Add in note.

3 Line 249-252, delete

4 I suggest author invite native English speaker to improve the writing English.

Author Response

Dear Reviewer,

               Thank you for your attention to the manuscript molecules-934375 entitle “N-way NIR data treatment through PARAFAC in the evaluation of antioxidants protective effect in soybean oil”.

The reviewers' comments were evaluated, and the suggestions were accepted (highlighted in blue on the text). We thank the reviewers for their suggestions and contributions. We hope that all these changes have contributed to improving the overall quality of the paper. We also hope that it now meets the publication standards in Molecules.

Thank you for considering this reviewed manuscript.

Sincerely,

Prof. Patrícia Valderrama

1 The method of ‘NIR spectroscopy in three-way arrays (samples x temperatures x absorbance at different wavelengths) by applying the PARAFAC chemometric tool to evaluate the protective effect over oxidation’ is interesting. Whether this method was applied in other components evaluation? Whether have other research reported this method? I suggest add the introduction of principle and application of methods in Part introduction.

We agree. A previous study by apply PARAFAC/NIR in the evaluation of rice oil through thermal degradation was included in the introduction part, where the principle of data organization was also added. We also introduce the previous works applying chemometrics and spectroscopy for edible oil oxidation studies in the discussion section, highlighting the PARAFAC applied to NIR data in the evaluation of rice oil thermal degradation.  While mathematical steps for the PARAFAC tool are cited in section 4.5 as detailed described by Bro [36].

2 In all figures, what is control? Add in note.

We agree.

3 Line 249-252, delete

We agree. We are sorry about this.

4 I suggest author invite native English speaker to improve the writing English.

We agree. The English were revised through the manuscript.